Transcriptome analysis revealed gene regulatory network involved in PEG-induced drought stress in Tartary buckwheat (Fagopyrum Tararicum)

Huang Juan
Chen Qijiao
Rong Yuping
Tang Bin
Zhu Liwei
Ren Rongrong
Shi Taoxiong
Chen Qingfu cqf1966@163.com
Guizhou Normal University, Research Center of Guizhou Buckwheat Engineering and Technology, Research Center of Buckwheat Industry Technology , Guiyang , Guizhou , P.R. China
Sun Genlou
Electronic publication date: 2021 Mar 31
Publication date: 2021
Volume: 9
Electronic Location ID: e11136
Received 2020 Oct 14; Accepted 2021 Mar 1
Copyright: ©2021 Huang et al.
Copyright year: 2021
Copyright holder: Huang et al.
License: This is an open access article distributed under the terms of the Creative Commons Attribution License, which permits unrestricted use, distribution, reproduction and adaptation in any medium and for any purpose provided that it is properly attributed. For attribution, the original author(s), title, publication source (PeerJ) and either DOI or URL of the article must be cited.
License URL: https://creativecommons.org/licenses/by/4.0/

Keywords: Drought stress, Transcriptome, ABA, Transcription factor, Tartary buckwheat

Funding: Joint project of National Science Foundation of China and Guizhou provincial government Karst Science Research Center U1812401 National Natural Science Foundation of China 31760419 32060508 Initial Fund for Doctor Research in Guizhou Normal University 11904/0516031 Guizhou Provincial Science and Technology Foundation QianKeHePingTaiRenCai [2017]5726 Natural Science Foundation of Guizhou Province QKHJC[2016]1106 Earmarked Fund for construction of the Key Laboratory for Conservation and Innovation of Buckwheat Germplasm in Guizhou QianJiaoHe KY Zi [2017]002 This work was supported by the Joint project of National Science Foundation of China and Guizhou provincial government Karst Science Research Center (U1812401), the National Natural Science Foundation of China (31760419, 32060508), and the Initial Fund for Doctor Research in Guizhou Normal University (11904/0516031), the Guizhou Provincial Science and Technology Foundation (QianKeHePingTaiRenCai [2017]5726), Natural Science Foundation of Guizhou Province (QKHJC[2016]1106), and the Earmarked Fund for construction of the Key Laboratory for Conservation and Innovation of Buckwheat Germplasm in Guizhou (QianJiaoHe KY Zi [2017]002). The funders had no role in study design, data collection and analysis, decision to publish, or preparation of the manuscript.

==============================
Tartary buckwheat is a nutritious pseudo-cereal crop that is resistant to abiotic stresses, such as drought. However, the buckwheat’s mechanisms for responding to drought stress remains unknown. We investigated the changes in physiology and gene expression under drought stress, which was simulated by treatment with polyethylene glycol (PEG). Five physiological indexes, namely MDA content, H2O2 content, CAT activity, SOD activity, and POD activity, were measured over time after 20% PEG treatment. All indexes showed dramatic changes in response to drought stress. A total of 1,190 differentially expressed genes (DEGs) were identified using RNA-seq and the most predominant were related to a number of stress-response genes and late embryogenesis abundant (LEA) proteins. DEGs were gathered into six clusters and were found to be involved in the ABA biosynthesis and signal pathway based on hierarchical clustering and GO and KEGG pathway enrichment. Transcription factors, such as NAC and bZIP, also took part in the response to drought stress. We determined an ABA-dependent and ABA-independent pathway in the regulation of drought stress in Tartary buckwheat. To the best of our knowledge, this is the first transcriptome analysis of drought stress in Tartary buckwheat, and our results provide a comprehensive gene regulatory network of this crop in response to drought stress.

Introduction

Tartary buckwheat (Fagopyrum Tararicum), also called bitter buckwheat, is a pseudo-cereal crop belonging to the genus Fagopyrum Mill, Polygonaceae (Ohnishi, 1998). Tartary buckwheat has become popular for its rich nutritional composition that includes a high content of flavonoids, resistant starches, crude fibers, proteins, and vitamins, which are all shown to have health benefits (Frias et al., 2011; Qin et al., 2010; Zhu, 2016). Tartary buckwheat is also highly adaptable to adverse soil and climatic conditions and shows a very strong tolerance or resistance to the adverse environment and abiotic stresses, including drought, low temperatures, and acid soils (Zhang et al., 2017).

Drought is a major meteorological disaster in agriculture of China and leads to the reduction of crop yield and quality (Xu et al., 2015). A series of physiological responses occur in Tartary buckwheat under water stress (Chen et al., 2008; Xiang et al., 2013), including a decrease of chlorophyll a, chlorophyll b, and total chlorophyll content and a significant increase in proline accumulation (Xiang et al., 2013). Photosynthesis, transpiration, stomatal conductance, and yield decrease but intercellular CO2 concentration increases (Xiang et al., 2013). Peroxidase (POD) activity, superoxide dismutase (SOD) activity, catalase (CAT) activity, hydrogen peroxide (H2O2) content, and soluble protein content, and proline content increase significantly and relative water content decreases rapidly in Tartary buckwheat under drought stress (Chen et al., 2008; Xiang et al., 2013).

Plants respond to water-deficit conditions with a series of processes at the physiological, cellular, and molecular levels to improve their stress tolerance (Shinozaki & Yamaguchi-Shinozaki, 2007). At the molecular level, the induction or suppression of candidate genes function to regulate plant damage and the stress response (Shinozaki & Yamaguchi-Shinozaki, 2007). Many drought-induced genes have been identified using large scale sequencing or gene function studies relating to functional proteins (transporters, detoxification enzymes, late embryogenesis abundant (LEA) proteins, proteases, key enzymes for osmolyte biosynthesis, and detoxification enzymes) and regulatory proteins (transcription factors (TFs), protein kinases, protein phosphatases, and proteins involved in signal biosynthesis and transduction, such as ABA biosynthesis and transduction) (Abdel-Ghany et al., 2020; Du et al., 2010; Frey et al., 2012; Li, Vallabhaneni & Wurtzel, 2008; Roychoudhury, Paul & Basu, 2013; Shi et al., 2020; Shinozaki & Yamaguchi-Shinozaki, 2007). Previous studies have supported that LEA proteins were involved in protecting higher plants from damage caused by environmental stresses, especially drought, by acting as antioxidants and membrane and protein stabilizers during water stress (Hong-Bo, Zong-Suo & Ming-An, 2005; Tunnacliffe & Wise, 2007). Genes encoding H2O2 scavengers, such as CAT, SOD and POD were also involved in the adaptation of plant drought responses (Luna et al., 2005; Molina-Rueda, Tsai & Kirby, 2013; Xiao et al., 2020). Many TFs, including ABRE2, RD29B, RD20A, MYB2, MBC2, RD26, DREB1D, HB, RD22, DREB2, NAC, and bZIP TFs (Fu et al., 2016; Lee et al., 2010; Roychoudhury, Paul & Basu, 2013; Shinozaki & Yamaguchi-Shinozaki, 2007; Yamaguchi-Shinozaki & Shinozaki, 2005) regulated plant drought resistance through two regulatory pathways: the ABA-dependent pathway and ABA-independent pathway (Abdel-Ghany et al., 2020; Du et al., 2010; Frey et al., 2012; Li, Vallabhaneni & Wurtzel, 2008; Roychoudhury, Paul & Basu, 2013; Shi et al., 2020; Shinozaki & Yamaguchi-Shinozaki, 2007).

Previous studies on Tartary buckwheat focused on changes in gene regulation under drought and identified some drought-inducible TFs, including three MYB family genes (FtMYB9, FtMYB10, and FtMYB13), eight NAC family genes (from FtNAC2 to FtNAC9), two bZIP family genes (FtbZIP5 and FtbZIP83), and a HLH family gene (FtbHLH3) (Deng et al., 2019; Gao et al., 2016; Gao et al., 2017; Huang et al., 2018; Li et al., 2019; Li et al., 2020; Sun et al., 2019; Yao et al., 2017). The transcriptome analysis of Tartary buckwheat under drought stress has not been reported despite the reports of other transcriptome analyses related to seed development (Huang et al., 2017; Liu et al., 2018). Many drought-inducible genes are unknown and the regulation mechanism of Tartary buckwheat under drought stress is still under investigation.

We examined the dynamic changes of five physiological indexes under drought stress, which was simulated by 20% polyethylene glycol (PEG-6000) treatment. We then performed a comprehensive transcriptome analysis using high throughput RNA-seq. The identification of differentially expressed genes (DEGs), the hierarchical cluster, GO enrichment, and KEGG enrichment was then analyzed. Genes in ABA biosynthesis and signal transduction were significantly enriched; thereafter ABA content after drought treatment was measured. The expression patterns of 18 important DEGs were verified using qRT-PCR. We proposed an ABA-dependent and ABA-independent pathway in the regulation of drought stress for Tartary buckwheat. This is the first known transcriptome analysis under drought stress for Tartary buckwheat, and provides a comprehensive gene regulatory network of Tartary buckwheat in response to drought stress.

Materials & Methods

Plant materials and growth conditions

A widely cultivated Tartary buckwheat variety, Jinqiao No. 2, was selected for its high stability and adaptability (Li et al., 2011). Seeds were surface sterilized with a 10% H2O2 solution, rinsed three times with double distilled water (ddH2O), and placed on wet filter papers for two days to accelerate germination. The uniformly sprouted seeds were moved to rolls of papers soaked with 0%, 10%, 20%, and 30% PEG-6000 solutions, respectively, to evaluate the genotypes after drought stress. After 48 h, the root length of the seedlings was compared across the four treatments. Three biological replicates were performed, and at least 7 seedlings were included in each biological replicate.

The uniformly sprouted seeds were moved to rolls of papers soaked with double distilled water when seedlings grew two true leaves, which occurred around day seven, in order to measure the physiological indexes and transcriptome sequencing. The seedlings with consistent growth were treated with 20% PEG based on the literature (Abdel-Ghany et al., 2020). Samples were taken at 0 h, 1 h, 3 h, and 6 h, frozen in liquid nitrogen, and stored at −80 °C. Three biological replicates were performed, and 40 seedlings were included in each biological replicate.

Physiological indexes measurement

A total of 5 physiological indexes, namely malondialdehyde (MDA) content, H2O2 content, CAT activity, SOD activity, and POD activity were measured, using determination kits (Sino Best Bio-Technology Co. Ltd, China) following the manufacturer’s instructions. Three biological replicates were included for each treatment, and 2 technical replicates were included for each biological replicate.

Total RNA isolation and transcriptome sequencing

Total RNA was isolated using the plant RNA purification kit (TianGen Biotech Co. LTD, China), according to the manufacturer’s instructions. The purified RNA samples were treated with Dnase I for 20 min to digest the genomic DNA. The quality and quantity of RNA were determined using the NanoDrop 2000 micro spectrophotometer, the Agilent 2100 Bioanalyzer, and the Agilent RNA 6000 Nano Kit. A total of 12 RNA samples were used to construct the library, and high-throughput sequencing was performed using the Illumina 4000 System (Illumina Inc., USA) with a read length of 150 bp and paired-end method.

Reads mapping and analysis of differentially expressed genes (DEGs)

Raw reads were filtered using Trimmomatic-0.38 (Bolger, Lohse & Usadel, 2014) with the following settings: trimmomatic-0.38.jar PE -threads 30 input_forward.fq.gz input_reverse.fq.gz output_forward_paired.fq.gz output_forward_unpaired.fq.gz output_reverse_paired.fq.gz output_reverse_unpaired.fq.gz LEADING:3 TRAILING:3 SLIDINGWINDOW:4:20 MINLEN:140. Quality control of all of the acquired clean reads was conducted using FastQC (https://github.com/s-andrews/FastQC) with the following parameters: fastqc -t 30 *.paired. Then the clean reads were mapped to the Tartary buckwheat genome data using Hisat2 (Kim, Langmead & Salzberg, 2015) with the following parameters: hisat2 -x genome -1 output_forward_paired -2 output_reverse_paired -S *.sam. The FPKM value for each gene was calculated using cufflinks (Trapnell et al., 2010). To evaluate the replicate reproducibility, PCA and Pearson correlation analyses were performed in R language (https://www.R-project.org/).

DEGs were identified with the DESeq2 package in R language, with the false discovery rate (padj) <0.05 and absolute value of Log2Ratio ≥ 1 as the thresholds. Gene annotation was performed by a local BLASP against the NR database, with the threshold of 1e−5. GO items were enriched by agriGO (http://bioinfo.cau.edu.cn/agriGO/) and the KEGG pathway was enriched by KOBAS 3.0 (http://kobas.cbi.pku.edu.cn/kobas3/?t=1). The significantly enriched GO items and KEGG pathways were visualized in R language.

Measurement of lycopene, zeaxanthin, and ABA content

High performance liquid chromatography was used to measure the content of lycopene, zeaxanthin, and ABA. For the measurement of lycopene and zeaxanthin, approximately 0.300 g well-ground samples were precisely weighed and placed in a 5 mL brown volumetric flask. Then 5 mL 0.1% bht-ethanol solution was added and oscillated for 5 min. The mixture was oscillated at 200 r/min for 4 h at room temperature in the dark. A 0.1% bht-ethanol solution was added to keep the volume of mixture at 10 mL. The mixture was subsequently centrifuged at 4,000 r/min for 10 min and 1 mL of the supernatant was filtered using a 0.22 µm Millipore filter, and the solution was collected in a 1.5 mL brown sample bottle. Lycopene and zeaxanthin were tested using Agilent HPLC-1100 with a DAD detector and Thermopylae C18 chromatographic column. The test conditions were as follows: column temperature, 25 °C; injection volume, 20 ul; flow rate, 1.0 ml/min; mobile phase, acetonitrile: methanol = 65:35 (V:V) isoelution.

Samples were well-ground and weighed to approximately 0.300 g to measure for ABA. Three mL were precooled and 80% methanol was added and mixed by oscillation. The mixture was sealed and stored overnight at 4 °C, then centrifuged at 5,000 r/min at 4 °C for 10 min. The supernatant was taken, and the residue was extracted ultrasonically with precooled 80% methanol at 4 °C for 30 min. This procedure was repeated twice and the supernatants were combined. The combined supernatant was blown to the aqueous phase with nitrogen at 4 °C. Three mL petroleum ether was added three times for decolorization, the aqueous phase was extracted with ethyl acetate three times then combined with the ethyl acetate phase and blown dry at 4 °C with nitrogen. The acetic acid solution (pH = 3.5) was added and purified in a SEP-PakC18 column. The eluent was eluted with methanol at room temperature and reduced to dry. ABA was tested by Agilent HPLC-1100 using the VWD detector and Agilent C18 chromatographic column (250*4.6 mm; 5 µL). The testing conditions were as follows: column temperature, 25 °C; injection volume, 10 ul; wavelength: 254 nm; flow rate, 1.0 ml/min; mobile phase, methanol: aqueous acetic acid solution (pH = 3.6) isoelution.

Quantitative RT-PCR (qRT-PCR) analysis

The transcriptome results were verified by qRT-PCR. A total of 31 DEGs were selected. Actin was used as the inner reference gene. Primer3Plus (http://www.primer3plus.com/cgi-bin/dev/primer3plus.cgi) was used to select gene-specific primers (Table S1). qRT-PCR was performed using the SYBR® Premix Ex Taq™ II kit (Takara Biomedical Technology (Beijing) Co., Ltd., China) on an ABI 7500 Fast Real-Time PCR system (Applied Biosystems, USA) following the manufacturer’s instructions, with three technical replicates. qRT-PCR results were calculated using the 2−ΔΔCt method.

Results

Investigation of drought tolerance of Tartary buckwheat

To determine the drought tolerance of Tartary buckwheat, we measured the root length of Tartary buckwheat seedlings after treatment with 0%, 10%, 20%, and 30% PEG. This method has been widely used to induce a water deficit in plants (Abdel-Ghany et al., 2020). The results are shown in Fig. 1. The seedlings treated with 10% PEG grew less root hair compared with those treated with 0% PEG; however, the root length showed no significant difference between the two treatments. The roots were dramatically shortened after treatment with 20% PEG compared to those treated with 0% PEG and 10% PEG. The roots did not grow well after treatment with 30% PEG, leading to the loss of approximately 50% of the seedlings. Seedlings treated with 20% PEG showed significant reductions in growth and root length, therefore, we used a 20% PEG treatment for the drought stress experiment.

Figure 1 Investigation of drought tolerance of Tartary buckwheat.

(A–D) The morphology of Tartary buckwheat seedlings treated by the PEG of 0%, 10%, 20%, 30%. (E) Root length quantification of Tartary buckwheat seedlings treated by the PEG of 0%, 10%, 20%, 30%.

Physiological changes of Tartary buckwheat seedlings under drought stress

We measured five physiological indexes involved in the drought stress response, namely MDA content, H2O2 content, CAT activity, SOD activity, and POD activity. These indexes were measured at four time points after being treated with 20% PEG and they showed significant alterations (Fig. 2). The H2O2 content was significantly increased after 1 h treatment, but remained nearly unchanged after 3 h and 6 h treatments. The MDA content was significantly increased after 1 h treatment and was significantly decreased afterwards. CAT activity showed an opposite trend to that of the H2O2 content, which was significantly decreased after 1 h treatment, but remained almost unchanged after 3 h and 6 h treatments. The SOD and POD activity showed an opposite trend to that of the MDA content, with a significant decrease after 1 h treatment, and a significant increase afterwards.

Figure 2 Physiological changes of Tartary buckwheat seedlings under drought stress.

(A) H2O2 content after PEG treatment. (B) MDA content after PEG treatment. (C) CAT activity after PEG treatment. (D) SOD activity after PEG treatment. (E) POD activity after PEG treatment.

High-throughput RNA-Seq of Tartary buckwheat seedlings under drought stress

We performed high-throughput RNA-Seq after 1, 3, and 6 h PEG treatments to obtain the transcriptome dynamics of Tartary buckwheat seedlings after drought tolerance. The 0 h treatment was the control. Three biological replicates were included for each treatment. We performed Pearson’s rank correlation analysis to evaluate the repeatability and reproducibility of the transcriptome data. The values of Pearson R between the two samples from the same biological replicates were higher than those of different biological replicates (Fig. S1), indicating that the biological replicates had good repeatability.

We obtained 44,303,640 to 62,033,328 raw reads and 42,564,428 to 59,376,546 clean reads for each library (Table 1). The quality of sequencing was high with the Q30 ranging from 92.53% to 93.29%. The genome data of Tartary buckwheat (Zhang et al., 2017) was used as the reference data for mapping, and we successfully mapped 74.49% to 75.45% of the clean reads to the predicted coding sequences (Table 1). A total of 27,490 genes were identified, with 24,433, 24,462, 24,508, 24,271, 24,680, 24,531, 24,617, 24,531, 24,398, 24,694, 24,553, and 24,263 genes identified in the libraries of PEG0h-1, PEG0h-2, PEG0h-3, PEG1h-1, PEG1h-2, PEG1h-3, PEG3h-1, PEG3h-2, PEG3h-3, PEG6h-1, PEG6h-2, and PEG6h-3, respectively (Table 1). We mapped 25,895, 25,982, 25,957, and 26,033 genes in the treatment of PEG0 h, PEG1 h, PEG3 h, and PEG6 h, respectively (Fig. 3A). We also performed a principal component analysis (PCA) and the results showed that samples were similar from the same time point after PEG treatment; whereas samples from the time point after the PEG treatment were not (Fig. 3B).

Table 1 Summary of RNA-Seq results under drought tolerance of Tartary buckwheat seedlings.

Sample	PEG0h-1	PEG0h-2	PEG0h-3	PEG1h-1	PEG1h-2	PEG1h-3	PEG3h-1	PEG3h-2	PEG3h-3	PEG6h-1	PEG6h-2	PEG6h-3	
Raw Reads Number	45,569,724	45,673,390	46,036,518	47,870,570	62,033,328	48,853,282	48,819,644	44,303,640	47,179,656	49,731,486	50,053,138	44,980,068	
Clean Reads Number	42,842,278	42,861,720	43,817,396	45,772,024	59,376,546	46,259,370	45,431,228	42,564,428	44,836,026	47,250,660	47,726,394	42,591,286	
Clean Q30 Bases Rate (%)	93.25	93.06	92.97	93.23	93.13	93.29	93.14	92.97	93.22	92.63	93.12	92.53	
Gene map Rate (%)	74.56	74.65	75.04	74.81	74.68	74.67	74.73	75.06	74.49	75.21	74.89	75.45	
Expressed Gene	24 433	24 462	24 508	24 271	24 680	24 531	24 617	24 531	24 398	24 694	24 553	24 263	

Figure 3 Global analysis of gene expression after 0 h, 1 h, 3 h, and 6 h treatment of PEG.

(A) the number of mapped genes in control and 3 treatments. (B) Principle component analysis (PCA) of all samples after PEG treatment.

Analysis of DEGs of Tartary buckwheat seedlings under drought stress

We analyzed DEGs using the pair-wise comparison of samples after different treatment times with 0 h PEG as the control to illustrate the transcriptome changes of Tartary buckwheat seedlings under drought stress. One thousand one hundred-ninety genes were up or down regulated by drought stress, among which 177, 558, and 315 DEGs were up-regulated in 1 h vs. 0 h, 3 h vs. 0 h, and 6 h vs. 0 h, whereas 36, 176, and 382 DEGs were down-regulated in 1 h vs. 0 h, 3 h vs. 0 h, and 6 h vs. 0 h (Fig. 4 and Table S2). Of these DEGs, 86 genes were co-up-regulated at 1 h vs. 0 h, 3 h vs. 0 h, and 6 h vs. 0 h, whereas 6 genes were co-down-regulated at 1 h vs. 0 h, 3 h vs. 0 h, and 6 h vs. 0 h. There were 53, 115, and 6 genes up-regulated in the comparisons of 1 h vs. 0 h, 3 h vs. 0 h, 3 h vs. 0 h, and 6 h vs. 0 h, and 1 h vs. 0 h and 6 h vs. 0 h, respectively, whereas 14, 77, and 0 genes were down-regulated in the comparisons of 1 h vs. 0 h and 3 h vs. 0 h, 3 h vs. 0 h and 6 h vs. 0 h, and 1 h vs. 0 h and 6 h vs. 0 h, respectively. The remaining genes were differentially regulated at one time point or showed no obvious expression patterns after PEG treatment (Fig. 4). It was worth to mention that circadian affect gene regulation. Previous study has suggested that more than one third expected DEGs were classified as clock-controlled genes comparing Arabidopsis sampled at time 0, 0.5, and 1 h (Hsu & Harmer, 2012). Thus, some fraction of the DEGs identified in our study might not be drought inducible genes, but circadian regulated genes.

Figure 4 Analysis of DEGs of Tartary buckwheat seedlings under drought stress.

Pink or dark blue bars represented the number of genes that were up or down regulated in comparison of 1 h, 3 h, and 6 h treatment 20% PEG with the control. Black bars represented the number of co-regulated genes under PEG treatment. Red or green dots represented genes that were up or down regulated in all three time point of PEG treatment. Orange or light blue dots represented genes that were up or down regulated in at least two time point of PEG treatment. Black dots represented genes that were differentially regulated in one time point or showed no obvious expression patterns after PEG treatment. URGs, up-regulated genes; DRGs, down-regulated genes.

The top 20 URGs and DRGs were analyzed after being exposed to drought stress for 1 h, 3 h, or 6 h (Tables 2, 3 and 4), among which many stress responsive genes were identified, including FtPinG0005419000.01 (annotated to dehydrin Rab18) and FtPinG0009412200.01 (annotated to carotenoid oxygenase). Interestingly, 5 LEA proteins (FtPinG0002083100.01, FtPinG0005679700.01, FtPinG0004425400.01, FtPinG0001202200.01, and FtPinG0000702400.01) were up-regulated at the third time point of the PEG treatments, suggesting the LEA proteins played crucial roles in response to drought stress. We found that that genes related to the oxidation–reduction process and reactive oxygen species biosynthesis were suppressed at the 1 h PEG treatment. These genes included two genes encoding peroxidase (FtPinG0007824100.01 and FtPinG0003282600.01) and one gene encoding allene oxide synthase (FtPinG0002376300.01), suggesting the biosynthesis of the protective enzyme, POD, was suppressed at a molecular level.

Table 2 List of top 20 URGs and DRGs by 1h PEG treatment.

Top 20 URGs	
Gene_ID	Gene annotation	Log2Ratio	FDR	
FtPinG0003580000.01	bidirectional sugar transporter N3	6.97	2.98E−03	
FtPinG0003652500.01	Embryonic protein DC-8	6.91	2.23E−21	
FtPinG0001202200.01	late embryogenesis abundant protein D-29	6.30	3.66E−05	
FtPinG0002083100.01	late embryogenesis abundant protein 2	5.68	1.81E−04	
FtPinG0000738700.01	uncharacterized protein	5.59	4.74E−06	
FtPinG0005768900.01	–	5.36	3.43E−02	
FtPinG0005419000.01	dehydrin Rab18	5.27	9.45E−15	
FtPinG0003707500.01	hypothetical protein	5.21	2.10E−03	
FtPinG0000339800.01	uncharacterized protein	4.89	1.48E−10	
FtPinG0000702400.01	late embryogenesis abundant protein D-29-like	4.55	8.81E−09	
FtPinG0004870600.01	low-temperature-induced 65 kDa protein	4.41	1.10E−19	
FtPinG0002722100.01	–	4.35	2.07E−03	
FtPinG0005679700.01	late embryogenesis abundant protein 46-like	4.34	3.83E−11	
FtPinG0008455800.01	translocator protein homolog	4.31	3.10E−07	
FtPinG0004425400.01	late embryogenesis abundant protein 6-like	4.29	1.19E−03	
FtPinG0006246100.01	uncharacterized protein	4.29	7.15E−06	
FtPinG0009412200.01	Carotenoid oxygenase	4.15	1.91E−05	
FtPinG0003146400.01	11-beta-hydroxysteroid dehydrogenase 1B-like	3.91	1.85E−03	
FtPinG0002481600.01	uncharacterized protein	3.83	6.35E−04	
FtPinG0009234700.01	cinnamoyl-CoA reductase 1	3.81	1.13E−02	
Top 20 DRGs	
Gene_ID	Gene annotation	Log2Ratio	FDR	
FtPinG0007824100.01	Peroxidase	−2.92	2.50E−02	
FtPinG0009798600.01	xyloglucan endotransglucosylase/hydrolase protein 23	−2.69	8.15E−13	
FtPinG0003282600.01	putative peroxidase N	−2.57	3.56E−03	
FtPinG0002337200.01	Thioredoxin-like	−2.34	2.00E−03	
FtPinG0008470300.01	hypothetical protein	−2.19	2.98E−03	
FtPinG0004263000.01	probable glycosyltransferase	−1.97	1.71E−02	
FtPinG0005512300.01	thioredoxin-like 1-2, chloroplastic	−1.88	4.90E−02	
FtPinG0006651300.01	sugar transporter ERD6-like 16	−1.86	3.65E−02	
FtPinG0005141700.01	ureide permease 2-like	−1.80	1.86E−02	
FtPinG0002376300.01	allene oxide synthase 1, chloroplastic	−1.79	1.95E−02	
FtPinG0003743200.01	indole-3-acetic acid-amido synthetase GH3.6	−1.71	3.24E−02	
FtPinG0006790900.01	beta-galactosidase 1	−1.69	2.98E−02	
FtPinG0008621600.01	protein trichome birefringence-like 41	−1.62	1.83E−03	
FtPinG0005799600.01	hypothetical protein	−1.57	2.45E−04	
FtPinG0004960100.01	F-box protein PP2-A12	−1.52	1.66E−02	
FtPinG0009594600.01	isocitrate lyase	−1.49	2.44E−02	
FtPinG0006479000.01	inositol oxygenase 1-like	−1.43	3.19E−02	
FtPinG0004575900.01	uncharacterized protein	−1.40	3.95E−02	
FtPinG0006731100.01	galactinol–sucrose galactosyltransferase 6	−1.39	7.80E−04	
FtPinG0003013100.01	uncharacterized protein	−1.39	6.45E−03	

Table 3 List of top 20 URGs and DRGs by 3 h PEG treatment.

Top 20 URGs	
Gene_ID	Gene annotation	Log2Ratio	FDR	
FtPinG0001894200.01	probable aldo-keto reductase 2	7.22	4.84E−09	
FtPinG0003652500.01	Embryonic protein DC-8	6.87	3.27E−07	
FtPinG0003580000.01	bidirectional sugar transporter N3	6.75	7.24E−15	
FtPinG0001202200.01	late embryogenesis abundant protein D-29	6.29	2.90E−07	
FtPinG0005846700.01	uncharacterized protein	5.80	7.65E−04	
FtPinG0002083100.01	late embryogenesis abundant protein 2	5.43	1.11E−06	
FtPinG0000738700.01	uncharacterized protein	5.38	3.38E−09	
FtPinG0001574200.01	germin-like protein subfamily 1 member 13	5.36	9.63E−46	
FtPinG0003983300.01	lipid transfer protein H	5.35	4.81E−02	
FtPinG0001106300.01	phenylpropene reductase 2	5.35	6.21E−10	
FtPinG0003707500.01	hypothetical protein	5.16	7.01E−05	
FtPinG0005419000.01	dehydrin Rab18	5.07	4.73E−09	
FtPinG0000702400.01	late embryogenesis abundant protein D-29-like	5.06	1.61E−03	
FtPinG0001100200.01	Reticulon-like protein	4.81	4.21E−02	
FtPinG0000339800.01	uncharacterized protein	4.76	1.92E−17	
FtPinG0005321900.01	aquaporin TIP3-1	4.71	1.84E−02	
FtPinG0004343900.01	membrane protein PM19L	4.65	8.39E−03	
FtPinG0009234700.01	cinnamoyl-CoA reductase 1	4.58	1.65E−20	
FtPinG0006246100.01	uncharacterized protein	4.54	1.97E−05	
FtPinG0005679700.01	late embryogenesis abundant protein 46-like	4.49	8.46E−16	
Top 20 DRGs	
Gene_ID	Gene annotation	Log2Ratio	FDR	
FtPinG0000545200.01	omega-hydroxypalmitate O-feruloyl transferase-like	−5.67	5.68E−03	
FtPinG0007033200.01	xyloglucan glycosyltransferase 4	−4.94	9.11E−04	
FtPinG0008272700.01	ABC transporter C family member 10 like	−3.90	9.18E−06	
FtPinG0006565000.01	root-specific metal transporter	−3.87	2.81E−02	
FtPinG0002543000.01	unknown	−3.81	1.08E−02	
FtPinG0007190100.01	Pectinesterase inhibitor domain protein	−3.70	9.13E−06	
FtPinG0006744800.01	protein TIFY 5A	−3.41	3.51E−02	
FtPinG0002671400.01	3-ketoacyl-CoA synthase 19-like	−3.20	1.21E−02	
FtPinG0007189700.01	Pectinesterase inhibitor domain protein	−2.90	5.42E−03	
FtPinG0009444400.01	cytochrome P450 724B1	−2.82	1.01E−02	
FtPinG0004261700.01	putative beta-D-xylosidase	−2.82	1.16E−07	
FtPinG0002765000.01	Methyltransferase PMT5	−2.82	8.12E−14	
FtPinG0009066900.01	alkane hydroxylase MAH1-like	−2.66	5.57E−05	
FtPinG0000019300.01	protein RADIALIS-like 5	−2.60	7.23E−06	
FtPinG0005398000.01	cytochrome P450 76AD1-like	−2.53	3.62E−07	
FtPinG0008470300.01	hypothetical protein	−2.49	2.95E−09	
FtPinG0004039900.01	pleiotropic drug resistance protein 3	−2.39	2.00E−02	
FtPinG0002454500.01	cytosolic sulfotransferase 15-like	−2.39	4.25E−02	
FtPinG0009798600.01	xyloglucan endotransglucosylase/hydrolase protein 23	−2.37	1.75E−11	
FtPinG0001540200.01	receptor-like protein kinase FERONIA	−2.35	1.75E−02	

Table 4 List of top 20 URGs and DRGs by 6 h PEG treatment.

Top 20 URGs	
Gene_ID	Gene annotation	Log2Ratio	FDR	
FtPinG0003983300.01	lipid transfer protein H	6.95	3.14E−02	
FtPinG0001202200.01	late embryogenesis abundant protein D-29	6.49	6.21E−06	
FtPinG0001894200.01	probable aldo-keto reductase 2	6.04	8.96E−06	
FtPinG0003652500.01	Embryonic protein DC-8	6.01	1.14E−04	
FtPinG0005846700.01	uncharacterized protein	5.59	2.63E−03	
FtPinG0005419000.01	dehydrin Rab18	5.59	6.66E−13	
FtPinG0002083100.01	late embryogenesis abundant protein 2	5.09	9.25E−06	
FtPinG0003146400.01	11-beta-hydroxysteroid dehydrogenase 1B-like	4.90	9.80E−03	
FtPinG0009234700.01	cinnamoyl-CoA reductase 1	4.89	4.45E−26	
FtPinG0005321900.01	aquaporin TIP3-1	4.71	1.70E−02	
FtPinG0003580000.01	bidirectional sugar transporter N3	4.60	9.39E−06	
FtPinG0005679700.01	late embryogenesis abundant protein 46-like	4.60	5.93E−24	
FtPinG0000702400.01	late embryogenesis abundant protein D-29-like	4.59	8.99E−03	
FtPinG0003707500.01	hypothetical protein	4.50	1.05E−02	
FtPinG0002481600.01	uncharacterized protein	4.45	7.07E−09	
FtPinG0006246100.01	uncharacterized protein	4.45	1.12E−04	
FtPinG0000835100.01	RACK1C	4.33	1.13E−04	
FtPinG0000339800.01	uncharacterized protein	4.18	2.90E−03	
FtPinG0000738700.01	uncharacterized protein	4.11	3.15E−04	
FtPinG0008455800.01	translocator protein homolog	3.96	1.18E−10	
Top 20 DRGs	
Gene_ID	Gene annotation	Log2Ratio	FDR	
FtPinG0004971800.01	methylesterase 10-like	−6.69	1.15E−04	
FtPinG0006470900.01	(3S,6E)-nerolidol synthase 2, chloroplastic/mitochondrial-like	−5.99	4.92E−04	
FtPinG0008375800.01	uncharacterized protein	−5.69	1.02E−02	
FtPinG0006565000.01	root-specific metal transporter	−5.41	9.38E−03	
FtPinG0001164200.01	bark storage protein A-like	−5.38	4.52E−02	
FtPinG0006744800.01	protein TIFY 5A	−4.87	1.14E−19	
FtPinG0007190100.01	Pectinesterase inhibitor domain protein	−4.56	1.59E−07	
FtPinG0005398000.01	cytochrome P450 76AD1-like	−4.35	1.87E−33	
FtPinG0004911500.01	lysine-specific demethylase JMJ30-like	−4.14	3.11E−06	
FtPinG0002221300.01	hypothetical protein	−4.02	4.64E−20	
FtPinG0001832800.01	2-isopropylmalate synthase A precursor	−3.46	1.25E−04	
FtPinG0002765000.01	Methyltransferase PMT5	−3.42	3.42E−19	
FtPinG0005904300.01	lysine-specific demethylase JMJ30-like	−3.30	1.73E−10	
FtPinG0006344900.01	zinc finger CCCH domain-containing protein 29	−3.26	1.39E−02	
FtPinG0005602100.01	hypothetical protein	−3.26	1.67E−02	
FtPinG0000412300.01	protein CUP-SHAPED COTYLEDON 3	−3.21	7.37E−03	
FtPinG0008416800.01	chaperone protein dnaJ C76, chloroplastic-like	−3.19	6.36E−23	
FtPinG0000385000.01	DNA-directed RNA polymerase II subunit RPB1 isoform X1	−3.15	6.98E−09	
FtPinG0001825500.01	protein FD-like	−3.10	4.12E−09	
FtPinG0009145200.01	ABC transporter B family member 21-like	−3.00	8.83E−03	

Functional category of DEGs of Tartary buckwheat under drought stress

We performed a hierarchical cluster analysis based on the gene expression pattern, with gene function enrichment based on GO annotation, to determine the biological function of the drought-responsive-genes. Six clusters (C1-C6) were identified (Fig. 5 and Table S2). C1 included 352 genes that were mildly down-regulated at 1 h and 3 h PEG, and were seriously down-regulated at 6 h PEG. A total of 11 GO biological processes were significantly enriched in this cluster, most of which were related to H2O2 metabolism and catabolism, reactive oxygen species metabolism, and oxidative stress responses. C2 included 189 genes that were up-regulated after PEG treatment; however, there was no significantly enriched GO biological process in this cluster. C3 included 87 genes that were down-regulated at 1 h and 3 h PEG, but up-regulated at 6 h PEG. A total of 14 GO biological processes were significantly enriched in this cluster, most of which were related to cell wall biogenesis, including xyloglucan metabolism, cell wall polysaccharide metabolism, and hemicellulose metabolism. C4 included 291 genes that were up-regulated at 1 h and 3 h PEG, but down-regulated at 6 h PEG, which showed an opposite expression pattern compared to C3. A total of four GO biological processes were significantly enriched in this cluster, and were related to lipid localization and phospholipid transport. C5 included 214 genes that were up-regulated at 1 h and 3 h PEG, but down-regulated at 6 h PEG, which showed a similar expression pattern to C4. A total of 25 GO biological processes were significantly enriched in this cluster, most of which were related to the regulation of candidate processes, such as regulation of gene expression and regulation of biosynthetic process. C6 included 57 genes that were down-regulated at 1 h PEG, but slightly up-regulated at 3 h and 6 h PEG treatment; however, there was no significantly enriched GO biological processes in this cluster.

We also analyzed the metabolic pathways based on KEGG annotation (Fig. 6). At 1 h PEG treatment, 200 of the 213 DEGS were annotated to 30 KEGG pathway, in which two pathways, “Carotenoid biosynthesis” and “Plant hormone signal transduction”, were significantly enriched (Qvalue <0.05, Fig. 6A). At 3 h PEG treatment, 702 of 734 DEGS were annotated to 81 KEGG pathways, in which six pathways were significantly enriched, namely “Glutathione metabolism”, “Protein processing in endoplasmic reticulum”, “alpha-Linolenic acid metabolism”, “Carotenoid biosynthesis”, “Biosynthesis of secondary metabolites”, and “Phenylpropanoid biosynthesis” (Qvalue <0.05, Fig. 6B). At 6 h PEG treatment, 634 of the 697 DEGS were annotated to 86 KEGG pathways, in which nine pathways were significantly enriched. The pathways were: “Biosynthesis of secondary metabolites”, “Circadian rhythm –plant”, “alpha-Linolenic acid metabolism”, “Phenylpropanoid biosynthesis”, “Metabolic pathways”, “Flavonoid biosynthesis”, “Protein processing in endoplasmic reticulum”, “Photosynthesis - antenna proteins”, and “Carotenoid biosynthesis” (Qvalue <0.05, Fig. 6C).

The involvement of ABA in response to drought stress in Tartary buckwheat

The carotenoid biosynthesis pathway was significantly enriched at all time points of PEG treatment, based on the KEGG results. Three DEGs were included in this pathway, namely two PSY genes (FtPinG0005737800.01 and FtPinG0004637900.01) and one BCH gene (FtPinG0006960700.01) (Fig. 7). PSY encodes phytoene synthase that catalyzes geranylgeranyl diphosphate to form phytoene, which is the rate-limiting enzyme in the carotenoid biosynthetic pathway. BCH encodes β-carotene hydroxylase of the P-450 monooxygenase family that converts β-carotene to zeaxanthin by a two-step reaction (Ruiz-Sola & Rodriguez-Concepcion, 2012). We determined the contents of phytoene and zeaxanthin to examine whether the carotenoids took a part in a drought response (Fig. S2). The result showed that neither phytoene nor zeaxanthin content were significantly altered after drought stress, indicating that the carotenoids may not be involved in drought response.

In higher plants, ABA is synthesized from the cleavage of carotenoid precursors (9′-cis-violaxanthin and 9′-cis-neoxanthin), which is likely the key regulatory step in the ABA biosynthetic pathway. The cleavage reaction is catalyzed by 9-cis-epoxycarotenoid dioxygenase (NCED), and produces xanthoxin, which can be converted into ABA via ABA-aldehyde (Chernys & Zeevaart, 2000). It is possible that ABA was involved in drought response rather than carotenoids. Interestingly, the plant hormone signal transduction KEGG pathway was one of the most enriched pathways in our results. Fourteen genes related to ABA biosynthesis and signal transduction were differentially expressed in our dataset (Fig. 7), including 4 DEGs homologous to NCED (FtPinG0009412200.01, FtPinG0006853200.01, FtPinG0003131500.01, and FtPinG0000246400.01), 2 DEGs homologous to AREB (FtPinG0002143600.01 and FtPinG0003196200.01), 5 DEGs homologous to PP2C (FtPinG0007629100.01, FtPinG0009574600.01, FtPinG0004850700.01, FtPinG0002889200.01, and FtPinG0006346100.01), 2 DEGs homologous to PYR (FtPinG0001214600.01 and FtPinG0007802700.01), and 1 DEG homologous to OST1 (FtPinG0003981600.01). Interestingly, 15 of the 17 DEGs showed highly consistent expression patterns and were classified to C5, with the exception of FtPSY2 (C4) and FtPYR1 (C1) (Table S2). Among these, two AREB homologs, FtPinG0002143600.01 and FtPinG0003196200.01, have been identified and named as FtbZIP83 and FtbZIP5 respectively in Tartary buckwheat previously (Li et al., 2019; Li et al., 2020).

Figure 5 Functional category of DEGs of Tartary buckwheat under drought stress.

(A) Hierarchical cluster of the drought-responsive-DEGs. (B–G) Expression patterns of the 6 clusters correspondent to the Hierarchical cluster result. Six main clusters were presented as C1–C6. (H) GO biological processes significantly enriched in six clusters, based on GO annotation. Missing GO-slim was represented by grey color.

Figure 6 Top 20 enrichment KEGG pathways under drought stress.

(A) Top 20 enrichment KEGG pathways at 1 h under drought stress. (B) Top 20 enrichment KEGG pathways at 3 h under drought stress. (C) Top 20 enrichment KEGG pathways at 6 h under drought stress.

Figure 7 ABA biosynthesis and signal transduction pathway were involved in response to drought stress.

(A) Overview of ABA biosynthesis and signal transduction pathway. The red rounded boxes represented substrates or products. The orange ellipse represented the enzymes or proteins. (B, C, D, F, G, H, and I) Heatmap represented the expression patterns of genes correspondent to the enzymes or proteins. The values were normalized to log2(FPKM). (E) ABA content after PEG treatment.

We measured the ABA content of the Tartary buckwheat seedlings after drought stress and the result of the physiological changes was in accordance with the molecular changes; the ABA content was significantly increased after PEG treatment (Fig. 7).

Transcription Factors (TFs) in response to drought stress in Tartary buckwheat

Two hundred fourteen DEGs in C5 were related to the regulation processes, such as regulation of gene expression and biosynthetic process, based on the hierarchical cluster and the enriched GO biological processes, which is indicative of TFs’ role in the drought stress response. We analyzed the differentially expressed TFs in the dataset. We found that a total of 174 TFs belonging to 36 TF families were identified as DEGs, among which the most abundant TF families were WRKY (27), NAC (14), MYB (13), C3H (12), bZIP (10), AP2-EREBP (8), DBP (7), HSF (7), Orphans (7), bHLH (6), C2H2 (6), and MYB-related (6) (Fig. 8). We then analyzed the expression patterns of the top 6 families and the genes in each TF family had a different expression pattern. Overall, most of the TFs were clustered to C5 (59 TFs), followed by C1 (38 TFs), C4 (32 TFs), and C2 (30 TFs) (Table S2), indicating their potential role on the regulation of definite biological processes. For example, among the 27 WRKY TFs, 10 TFs were clustered to C5, suggesting that they may function in the regulation of biosynthetic process; seven TFs were clustered to C4, suggesting they may function in lipid localization and phospholipid transport; three TFs were clustered to C1, suggesting that they may function in H2O2 metabolism and catabolism, reactive oxygen species metabolism, and oxidative stress response; and three TFs were clustered to C3, suggesting they may function in cell wall biogenesis (Figs. 5 and 8). In addition, some TFs, such as FtPinG0002173200.01, FtbZIP83, FtbZIP5, FtPinG0007618600.01, and FtPinG0008274300.01, were homologous to the reported genes within the regulatory network of the drought stress response (Fujita et al., 2004; Lee et al., 2010; Li, Vallabhaneni & Wurtzel, 2008; Li et al., 2019; Li et al., 2020; Trivedi, Gill & Tuteja, 2016).

Figure 8 Summary and expression patterns of TFs in response to drought stress in Tartary buckwheat.

(A) Statistics of the identified TFs. (B–G) Expression patterns of the large families of NAC (B), WRKY (C), AP2-EREBP (D), MYB (E), C3H (F), and bZIP (G). The values were normalized to log2(FPKM).

Confirmation of the transcriptome data by qRT-PCR

The expression patterns of 31 DEGs were verified by qRT-PCR, including five LEA protein encoding genes, seven genes in ABA biosynthesis (FtNCED1, FtNCED2, FtNCED3, FtNCED4, FtPSY1, FtPSY2, and FtBCH), 16 genes in ABA signal pathway (FtABRE1, FtABRE2, FtPP2C1, FtPP2C2, FtPP2C3, FtPP2C4, FtPP2C5, FtPYR1, FtPYR2, FtOST1, FtRD29B, FtRD26, FtDREB1D, FtHB1, FtHB2, and FtRD22), and three TFs in the ABA-independent pathway (FtDREB2, FtRD19, and FtERD1). The results showed that the expression patterns obtained by qRT-PCR were highly correlated with those obtained by transcriptome data, with a Pearson R correlation ranging from 0.76 to 0.98 (Fig. 9), suggesting that our transcriptome results were reliable.

Figure 9 Confirmation of the transcriptome data by qRT-PCR.

(A) FtPSY1. (B) FtPSY2. (C) FtBCH. (D) FtNCED1. (E) FtNCED2. (F) FtNCED3. (G) FtNCED4. (H) FtPYR1. (I) FtPYR2. (J) FtPP2C1. (K) FtPP2C2. (L) FtPP2C3. (M) FtPP2C4. (N) FtPP2C5. (O) FtOST1. (P) FtbZIP83. (Q) FtbZIP5. (R) FtLEA1. (S) FtLEA2. (T) FtLEA3. (U) FtLEA4. (V) FtLEA5. (W) FtRD29B. (X) FtRD26. (Y) FtDREB1D. (Z) FtHB1. (AA) FtHB2. (BB) FtRD22. (CC) FtDREB2. (DD) FtRD19. (EE) FtERD1.

Discussion

Physiological changes were highly consistent with molecular changes of Tartary buckwheat seedlings under drought stress

ROS accumulate in high amounts when a plant survives oxidative damage or various stresses, such as drought, leading to the damage of the membrane protein and lipids. H2O2 and MDA content increase rapidly under these conditions and the ROS scavenger enzymatic system, including the activities of SOD, APX, and CAT, are induced as an universal response (Mohammadkhani & Heidari, 2007). We measured five widely used physiological indexes in drought stress response, namely MDA content, H2O2 content, CAT activity, SOD activity, and POD activity, after 20% PEG treatment. The results showed that the physiological changes were highly consistent with the molecular changes based on the hierarchical cluster and GO enrichment (biological process) analyses (Figs. 2 and 5, and Table S2). The H2O2 and MDA content were significantly increased in the 1 h PEG treatment, whereas the activity of the protective mechanisms was decreased. Genes related to H2O2 metabolism and catabolism, and ROS metabolism, were clustered to C1 and were down-regulated at this time point, suggesting that ROS accumulation and membrane damage immediately occurred after drought stress in Tartary buckwheat. In the 3 h and 6 h PEG treatment, the activity of the protective enzymes significantly increased, but the MDA content decreased, indicating that the ROS scavenger system was induced against drought stress. The related genes were clustered to C3 and C4, which were associated with cell wall biogenesis and metabolism and lipid localization and phospholipid transport. This suggests that the repair of the ROS and damage to the membrane occurred after the damage to the ROS and membrane.

LEA proteins were involved in drought stress response in Tartary buckwheat

LEA proteins are a type of low molecular weight protein induced by various abiotic stresses, such as drought, high temperature, and cold (Hong-Bo, Zong-Suo & Ming-An, 2005). A number of studies have reported that genes encoding LEA proteins are induced to maintain the stability of the membranes and proteins and to provide detoxification and alleviation of cellular damage under conditions of dehydration (Roychoudhury, Paul & Basu, 2013; Shi et al., 2020; Shinozaki & Yamaguchi-Shinozaki, 2007; Tunnacliffe & Wise, 2007). In a recent transcriptome comparison of drought-resistant and drought-sensitive sorghum genotypes, six of the 25 top-induced genes in drought-resistant genotypes encoded LEA proteins (Abdel-Ghany et al., 2020). Our results were similar (Tables 2, 3 and 4). Twenty-nine up-regulated genes were identified, out of which five genes were annotated to LEA proteins. Among these, four genes were up-regulated at all time points after PEG treatment, including two genes homologues to LEA protein D-29, one gene homologous to LEA protein 2, and one gene homologous to LEA protein 46. Their expression levels were high, with an absolute fold change of 20 to 90 times, suggesting that LEA proteins were involved in the drought stress response in Tartary buckwheat as well.

Transcriptional regulatory network in ABA-dependent and ABA-independent pathways under drought stress of Tartary buckwheat

Plants have two regulatory pathways, the ABA-dependent pathway and the ABA-independent pathway, to manage drought responsive genes (Roychoudhury, Paul & Basu, 2013; Shinozaki & Yamaguchi-Shinozaki, 2007). To better understand the ABA-dependent and ABA-independent pathways of Tartary buckwheat under drought stress, a transcriptional regulatory network was constructed based on the previously reported regulatory network and data obtained from our study (Fig. 10).

Figure 10 Transcriptional regulatory network in ABA-dependent and ABA-independent pathways under drought stress of Tartary buckwheat.

Modified from previous studies (Fu et al., 2016; Roychoudhury, Paul & Basu, 2013; Shinozaki & Yamaguchi-Shinozaki, 2007). Blue and pink rounded boxes represent genes in the ABA-dependent and ABA-independent pathway, respectively. Rounded boxes with no border indicate genes were consistent with the reported studies, whereas rounded boxes with red dotted lines indicate genes reported in previous studies were not identified in our study.

The ABA-dependent pathway under drought stress begins with ABA biosynthesis. ABA biosynthesis begins from the cleavage of carotenoid precursors catalyzed by NCED, which is the key regulatory step in the ABA biosynthetic pathway (Chernys & Zeevaart, 2000; Seo & Koshiba, 2002). We found that four genes (FtNCED1, FtNCED2, FtNCED3, and FtNCED4) encoding NCED in ABA biosynthesis and three genes (FtPSY1, FtPSY2, and FtBCH) encoding key enzymes (PSY and BCH) in carotenoid biosynthesis were up-regulated (Fig. 7). All of these genes were found to be induced by drought stress or other abiotic stresses in previous studies (Du et al., 2010; Frey et al., 2012; Li, Vallabhaneni & Wurtzel, 2008). There are at least five signal pathways that regulate drought stress in the ABA-dependent pathway in plants. The primary pathway is the ABRE2 mediated signal pathway (Roychoudhury, Paul & Basu, 2013; Shinozaki & Yamaguchi-Shinozaki, 2007). We identified two genes homologous to ABRE2 (FtbZIP83 and FtbZIP5), and eight genes (FtPYR1, FtPYR2, FtPP2C1, FtPP2C2, FtPP2C3, FtPP2C4, FtPP2C5, and FtOST1) in the upstream of ABRE2 as DEGs (Fig. 7). Among these, FtbZIP83 and FtbZIP5 were reported to improve drought/salt tolerance via an ABA-mediated pathway in Transgenic Arabidopsis recently (Li et al., 2019; Li et al., 2020). ABRE2 binds to the DRE/CRT element in the promoter of RD29B and RD20A to activate their expression in response to stress (Yamaguchi-Shinozaki & Shinozaki, 2005). We identified a gene (FtPinG0004870600.01) homologous to RD29B that was induced under drought stress in Tartary buckwheat, however, the homolog of RD20A was not among the identified DEGs (Fig. 10). The ABRE2 mediated signal pathway in addition to the MYB2 and MBC2, RD26, DREB1D, and HB mediated signal pathways are also important constituents in plant ABA-dependent pathway (Roychoudhury, Paul & Basu, 2013; Shinozaki & Yamaguchi-Shinozaki, 2007). We found four homologs of RD26 (FtPinG0002173200.01), DREB1D (FtPinG0008274300.01), and HB (FtPinG0001748800.01 and FtPinG0008157500.01), all of which were up-regulated under drought stress (Table S2). Though we found no homologs of MYB2 or MBC2, one gene (FtPinG0002802300.01) homologous to its downstream gene, RD22, was identified in our data. It was down-regulated after drought stress and exhibited a different expression pattern compared to other genes in the ABA-dependent pathway (Table S2 and Fig. 5). Sixteen of the 20 DEGs mentioned above showed similar expression patterns and were up-regulated at 1 h and 3 h PEG, but down-regulated at 6 h PEG, and clustered to C5 (Table S2, Fig. 5 and Fig. 7), with the exception of FtPSY2, FtPYR1, FtPinG0008274300.01 and FtPinG0002802300.01. These results confirmed that genes in ABA-dependent regulatory pathway are co-regulators in response to drought stress in Tartary buckwheat. In addition, changes of ABA content corresponded with the molecular changes, and confirmed the role of ABA in response to drought stress and tolerance in buckwheat (Fig. 7).

There are at least two signal pathways, namely the DREB2 mediated signal pathway and the NAC and bZIP TFs mediated signal pathway, to regulate drought stress in the plant ABA-independent pathway (Fu et al., 2016; Roychoudhury, Paul & Basu, 2013; Shinozaki & Yamaguchi-Shinozaki, 2007). DREB2 belongs to the AP2-EREBP TF family and may transcriptionally activate the expression of RD19 and RD29A (Lee et al., 2010). We identified the homologues of DREB2 (FtPinG0007618600.01) and RD19 (FtPinG0002253700.01) but were unable to identify the homolog of RD29A (Table S2 and Fig. 10). Massive NAC and bZIP family TFs have been reported to function in the ABA-independent pathway by activating ERD1, which encodes a chloroplast-targeted Clp protease regulatory subunit that is induced by water stress in Arabidopsis thaliana (Nakashima et al., 2010). We identified a homolog of ERD1 (FtPinG0001990100.01) which was clustered to C2 based on its expression pattern (Table S2 and Fig. 5). Interestingly, some NAC and bZIP TFs were also clustered to C2, suggesting they may be co-expressed and in the upstream of ERD1 (Table S2, Figs. 5, 8 and 10).

The ABA-dependent and ABA-independent pathways were found to participate in the regulation of drought stress of Tartary buckwheat.

Conclusion

We investigated the physiological changes and the gene expression changes in a time-course manner under drought stress simulated by 20% PEG treatment. A total of 1,190 DEGs were identified and the genes encoding LEA proteins were listed on the top up-regulated DEGs. All DEGs were grouped into six clusters, in which genes showed definite expression patterns and were involved in specific biological processes based on GO annotation. Further analyses of the ABA and TFs revealed they were also involved in the drought stress response in Tartary buckwheat. We proposed ABA-dependent and ABA-independent pathways in the regulation of drought stress of Tartary buckwheat. This is the first study using a large-scale sequencing method to unravel the transcriptomic changes under drought stress in Tartary buckwheat, which identified massive genes and a gene regulatory network in response to drought stress. Our study provides candidate genes for further functional studies.

Supplemental Information

Figure S1 Pearson’s rank correlation analysis of all samples after PEG treatment

Click here for additional data file.

Figure S2 The contents of phytoene and zeaxanthin under drought stress

Click here for additional data file.

Table S1 List of qRT-PCR primers

Click here for additional data file.

Table S2 DEGs identified under drought stress

Click here for additional data file.

Supplemental Information 5 Raw data of Fig. 1B

Click here for additional data file.

Supplemental Information 6 Raw data of Fig. 2

Click here for additional data file.

Supplemental Information 7 Raw data of Fig. 7

Click here for additional data file.

Supplemental Information 8 Raw data of Fig. 9

Click here for additional data file.

Additional Information and Declarations

Competing Interests

Author Contributions

DNA Deposition

Data Availability

The authors declare there are no competing interests.

Juan Huang conceived and designed the experiments, performed the experiments, analyzed the data, prepared figures and/or tables, authored or reviewed drafts of the paper, and approved the final draft.

Qijiao Chen performed the experiments, analyzed the data, prepared figures and/or tables, authored or reviewed drafts of the paper, and approved the final draft.

Yuping Rong performed the experiments, analyzed the data, prepared figures and/or tables, and approved the final draft.

Bin Tang and Liwei Zhu and Rongrong Ren performed the experiments, prepared figures and/or tables, and approved the final draft.

Taoxiong Shi conceived and designed the experiments, prepared figures and/or tables, and approved the final draft.

Qingfu Chen conceived and designed the experiments, authored or reviewed drafts of the paper, and approved the final draft.

The following information was supplied regarding the deposition of DNA sequences:

Raw data are available at the China National Center for Bioinformation Genome Sequence Archive: CRA003335, PRJCA003569.

The following information was supplied regarding data availability:

Raw measurements used to create Figs. 1B, 2, 7, and 9 are available in the Supplemental Files.

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
