# Peer review of "Transcriptome analysis revealed gene regulatory network involved in PEG-induced drought stress in Tartary buckwheat (Fagopyrum Tararicum)"

_PeerJ, doi:10.7717/peerj.11136_

## Round 0.1 · original submission · Major Revisions

We have the reviewers' comments on your manuscript. please revise it according to the comments.

Reviewer 1 ·

Basic reporting

no comment

Experimental design

no comment

Validity of the findings

no comment

Additional comments

In this manuscript, authors presented some work regarding Transcriptome analysis revealed gene regulatory network involved in PEG-induced drought stress in tartary buckwheat (Fagopyrum Tararicum Gaerth). There are the following important issues in the manuscript.
1. Abstract
The abstract must be self-contained and concisely describe the reason for the work, methodology, results, and conclusions.
2. Introduction:
Lines 46-50: You should provide the literature you cite.
Lines 57-61: Do you want to express some drought induced genes in Tartary buckwheat have been identified. You should provide the literature you cite.
Your introduction needs more detail. I suggest that you improve the whole description to provide more justification for your study (Have the physiological and transcriptional changes of Tartary Buckwheat under drought stress been explored (Lines 55-57)? Whether the transcriptome analysis of Tartary Buckwheat under drought stress has been reported? Is there any difference between this study and the previous report? Why should the author do this research again?).
In addition, the author did not elaborate the present research progress of Tartary buckwheat clearly. Many important transcription factor families that regulate growth and development and resist environmental stress have been identified in Tartary buckwheat, including AP2/ERF, BHLH, NAC, ARF, et al. Meanwhile, transcriptome analysis related to Tartary buckwheat development have also been reported (Liu et al. Insights into the correlation between Physiological changes in and seed development of tartary buckwheat (Fagopyrum tataricum Gaertn.). BMC Genomics (2018) 19: 648). The public information would be helpful identifying important candidate genes.
The author does not clarify the scientific problems of this study in the introduction. The introduction part is not well written and needs to be rearranged.
Lines 20, 38, 40, and many more, including lines 174, 185 and so on. Instead of »tartary buckwheat« correct to »Tartary buckwheat«, where applicable. In difference to »common buckwheat« (with lower-case first letter), »Tartary buckwheat« should be in English always written by upper-case first letter; this way of writing was internationally accepted, among others in the discussion on International Buckwheat Research Association (IBRA) Assembly in 2013, as word »Tartary« comes from the name of the Tartar people.
3. Methods:
Line 94: Why choose these four time points (0 h, 1 h, 3 h, and 6 h)?
Whether the author uploaded the raw data?
4. Results:
Lines 185-195: The descriptive results of physiological indexes after drought stress were provided. What readers want to know is how to explain the fluctuation of these physiological indexes and whether there is a reasonable explanation. Is there any change in the phenotype of Tartary buckwheat after drought treatment for different time?
Lines 213-215 and so on: Spaces are required between numbers and units.
Line 307: Should it be NECD or NCED?
5. Figures:
In Figure 2 and figure 3, there should be a space between the number and the unit in the abscissa
Figures 1, 2, 8, and 9 have multiple parts. Each figure with multiple parts should have alphabetical (e.g. A, B, C) labels on each part and all parts of each single figure should be submitted together in one file. In this case:
The 5 parts of Figure 1 should be labeled A-E.
The 5 parts of Figure 2 should be labeled A-E.
The 18 parts of Figure 8 should be labeled A-R.
The 7 parts of Figure 9 should be labeled A-G.
Please provide replacement figures measuring minimum 900 pixels and maximum 3000 pixels on all sides, saved as PNG, EPS, or PDF (vector images only) file format without excess white space around the images.
Text and figures legends should also be adjusted appropriately
6. References:
Please check the style of references. There are many references with incomplete formatting and missing page numbers, for example, “Abdel-Ghany SE, Ullah F, Ben-Hur A, and Reddy ASN. 2020. Transcriptome Analysis of Drought-Resistant and Drought-Sensitive Sorghum (Sorghum bicolor) Genotypes in Response to PEG-Induced Drought Stress. Int J Mol Sci 21. 10.3390/ijms21030772”.
7. Language
The English language should be improved to ensure that an international audience can clearly understand your text. Some examples where the language could be improved include lines 199, 174-176, 177– the current phrasing makes comprehension difficult.

Reviewer 2 ·

Basic reporting

no comment

Experimental design

no comment

Validity of the findings

no comment

Additional comments

1.English language and style. Moderate English changes required .
2.Introduction part must be significantly improved, some background of candidate genes should be added, for example: what is the candidate genes and why it needs to identify etc.
3.Are the methods adequately described?
4.Is the research design appropriate?
5.Why choose Jinqiaomai2 as the material?
6.The discussion part must be improved. In this study, the relationship between physiological changes and transcriptome data, and changes in gene expression under drought stress?
7.Supplement LEA protein expression experiment.
8. Authors must be highlighted the significant results found in this study as a conclusion.

Reviewer 3 ·

Basic reporting

Drought is a major environmental factor can influence yield and quality of crop. The work used the transcriptome to analyze the effects of drought on gene expression in Tartary buckwheat. The overall design and experiment of this paper are relatively complete. But there are some small questions that need to be answered.

Experimental design

This study design is accord with the standard experimental design of transcriptom.

Validity of the findings

1. Line 251, sight maybe wrong, should be slight.
2. Finally, the transcription factors selected by you may be involved in Tartary buckwheat drought stress. You should verify the expression of these transcription factors again by qRT-PCR.
3. In the third paragraph of the results you don't seem to have written about gene expression, so the subtitle might leave out the “global analysis of gene expression”.

---

## Round 0.2 · Minor Revisions

A Section Editor comments on your manuscript as follows:

"There is a serious design flaw. Zero hours is used as the control for 1, 3, and 6 hour PEG treatments. The problem with this design is that at least one third of all genes are regulated in a circadian fashion. This means that even with no treatment, comparing samples collected 6 hours apart would yield 100s to 1000s of differentially expressed genes. Please see figure 7 at https://journals.plos.org/plosone/article?id=10.1371/journal.pone.0049853 . Thus, some (potentially large) fraction of the genes identified here as being responsive to drought are just differentially expressed because of the difference in collection time.

The correct way to design the experiment would be to have a no-treatment control collected at every time point. At a minimum the manuscript needs to modified to acknowledge this shortcoming and that many of the genes may not actually be related to drought stress.

Better, of course, would be to do the experiment correctly but I won't require that so long as the caveats of this design are adequately explained."

Please address this issue in your next revision,

Please add several sentences to acknowledge the design shortcoming and that many of the genes may not actually be related to drought stress.

Reviewer 1 ·

Basic reporting

no comment

Experimental design

no comment

Validity of the findings

no comment

Additional comments

no comment

Reviewer 2 ·

Basic reporting

no comment

Experimental design

no comment

Validity of the findings

no comment

Additional comments

The authors have adequately addressed all of my comments and suggestions.

Reviewer 3 ·

Basic reporting

The revised manuscript satisfied with my suggestion and reach the requirement of acceptance.

Experimental design

The revised manuscript satisfied with my suggestion and reach the requirement of acceptance.

Validity of the findings

The revised manuscript satisfied with my suggestion and reach the requirement of acceptance.

Additional comments

The revised manuscript satisfied with my suggestion and reach the requirement of acceptance.

---

## Round 0.3 · accepted · Accept

The authors have revised the manuscript according to the comments.